# Effect of Pulsed Electric Field (PEF) on Bacterial Viability and Whey Protein in the Processing of Raw Milk

Aivaras Šalaševičius, Dovilė Uždavinytė, Mindaugas Visockis, Paulius Ruzgys and Saulius Šatkauskas *

Biophysical Research Group, Faculty of Natural Sciences, Vytautas Magnus University, Vileikos St. 8, LT-44404 Kaunas, Lithuania; aivaras.salasevicius@vdu.lt (A.Š.); dovile.uzdavinyte@vdu.lt (D.U.); mindaugas.visockis@lammc.lt (M.V.); paulius.ruzgys@vdu.lt (P.R.)
* Correspondence: saulius.satkauskas@vdu.lt

**Abstract:** There is growing concern regarding the nutritional value of processed food products. Although thermal pasteurization, used in food processing, is a safe method and is widely applied in the food industry, food products lack quality and nutritional value because of the high temperatures used during pasteurization. In this study, the effect of pulsed electric field (PEF) processing on whey protein content and bacterial viability in raw milk was evaluated by changing the PEF strength and number of pulses. For comparison, traditional pasteurization techniques, such as low-temperature long-time (LTLT), ultra-high temperature (UHT), and microfiltration (MF), were also tested for total whey protein content, bacterial activity, and coliforms. We found that, after treatment with PEF, a significant decrease in total bacterial viability of 2.43 log and coliforms of 0.9 log was achieved, although undenatured whey protein content was not affected at 4.98 mg/mL. While traditional pasteurization techniques showed total bacterial inactivation, they were detrimental for whey protein content: β-lactoglobulin was not detected using HPLC in samples treated with UHT. LTLT treatment led to a significant decrease of 75% in β-lactoglobulin concentration; β-lactoglobulin content in milk samples treated with MF was the lowest compared to LTLT and UHT pasteurization, and ~10% and 27% reduction was observed.

**Keywords:** milk; PEF; food nutrition; bacterial inactivation; whey protein; microbiological safety

## 1. Introduction

### 1.1. Application of Pulsed Electric Fields (PEF) in Food Processing and Pasteurization

Thermal pasteurization of food products and beverages has specific drawbacks. During the process of thermal pasteurization, a fraction or the whole nutritional value of a product is lost because of the high temperatures applied to constituents such as vitamins and proteins found in raw food [1–3]. An inability to ensure daily intake of vitamins, proteins, carbohydrates, electrolytes, and fats leads to specific forms of malnutrition. Therefore, malnutrition can lead to other diseases, such as kwashiorkor, marasmus, or anemia [4]. The growing consumer demand for fresh, low-processed, and healthy foods has also prompted the food industry to look for alternatives to traditional pasteurization methods. Pulsed electric field (PEF) technology is an innovative method that has been adopted in the food industry. The first mention of juice and milk being pasteurized dates back to 1989–1990s, performed by the Krupp Industrietechnik GmbH company [5]. PEF processing technology has been widely adopted over the years and has been successfully used in fields such as medicine, food processing, and bio-based industries [6–8]. It has a wide range of applications in medicine for drug and gene delivery and in cancer research and laboratory settings for cell fusion [9–11]. In the food industry, such as juice and potato fry processing, it is shown to ensure higher juice yields and better quality of potato chips [12–14]. PEF is being applied in mild liquid food and beverage pasteurization [15–17].

The applicability of PEFs has been discussed in several studies [17–21]. The main advantage of PEF technology in liquid food pasteurization is the ability to manage the amount of ohmic heating in food preservation (low-temperature processing), avoiding the Maillard reaction, which affects the functional properties of food, such as color, taste, and smell [22]. PEF is also effective in the inactivation of microorganisms such as *Salmonella typhimurium*, *Listeria innocua*, and *E. coli* up to 5.0 log cycles [23]. The method is highly scalable and can be incorporated into existing food processing lines. In comparison to traditional heat pasteurization technology, it is more energy-efficient [6,24]. Furthermore, PEF treatment chambers can be easily adapted to existing continuous-flow production lines for liquid food pasteurization [25]; however, achieving a homogeneous treatment may be an issue [26]. The main disadvantage of PEF technology is its effectiveness and efficiency, which are largely dependent on the liquid conductivity and viscosity [27]. Dielectric breakdown and non-uniform treatment may occur because of the existing air or gas bubbles [28]. It has been shown that electrode corrosion can be an issue during PEF as small amounts of electrode material may reside in the liquid being pasteurized [29,30]. Another issue is bacterial spores, which are highly resistant to PEF technology, because of the spore outer coat and cortex; therefore, PEF can only be considered as a pasteurization technique and cannot be used for sterilization [31–33]. Nonetheless, liquid food sterilization by combining PEF and pressurized flow systems is gaining ground and is being tested in laboratory settings [34].

### 1.2. Nutritional Properties of Raw Whey Proteins

Raw whey proteins found in milk are highly prized owing to their nutritional value, fast absorption, and high levels of antioxidants, amino acids, and peptides [35]. Whey protein (WP) preparations can be divided into three categories: protein isolates, concentrates, and hydrolysates [36,37]. Whey protein isolates and concentrates are highly adopted in the sports industry, where they help to enhance whole-body protein metabolism and skeletal muscle growth [36,38]. Whey proteins have been shown to increase satiety and suppress the feeling of hunger, thereby reducing short-term food intake [39,40]. Studies have reported that WP can reduce postprandial glycemia and have a glucose-lowering effect [41,42]. People affected by Phenylketonuria must adhere to daily nutritional plans free of phenylalanine amino acid. In these cases, whey protein glycomacropeptide can be a good substitute and can serve as a new food supplement for people suffering from Phenylketonuria [43]. Studies on whey-protein-enriched food production and their nutritional value have recently gained increased interest. It has been reported that adding erythritol in the production of WP isolate meringues can help to improve the overall structural properties of the final product [44]. Additionally, yoghurts enhanced with WP isolates have been shown to possess improved textural properties, as well as higher nutritive value [45].

Whey proteins mostly consist of β-lactoglobulin, α-lactalbumin, glycomacropeptide, immunoglobulins, bovine serum albumin, lactoferrin, and lactoperoxidase [46,47]. It has been reported that β-lactoglobulin has a high concentration of cysteine and other amino acids, which stimulate the production of glutathione in mice for protection against intestinal tumors [48]. Lactose synthesis is supported by α-lactalbumin, which is the main energy source for newborn children [49]. However, due to the high temperatures used during the pasteurization of raw milk, whey protein content is affected [50]. This is a huge issue in developing countries, where the daily intake of food products and their nutritional content are low. Malnutrition in the long term leads to undernourishment and hunger.

PEF can bridge the gap between safe food processing and products with high nutritional value. However, there is still much to be done to achieve complete bacterial inactivation using PEF in liquid food pasteurization. It is a known fact that the efficacy of the microbial inactivation by PEF mainly depends on the proper selection of technological parameters such as PEF strength, treatment duration, specific energy input, pulse repetition frequency, conductivity, temperature, pH, and the shape and composition of the electrodes [8,51–53]. However, PEF technology has many variables that differ from one

liquid to another; therefore, finding the optimal parameters for these variables and their compositions might be challenging.

The working hypothesis of this study was that PEF can lead to efficient milk pasteurization while preserving nutrient value. Hence, the aim of this study was to examine the effect of PEF on whey protein content and bacterial inactivation compared to traditional pasteurization techniques used in the food industry.

## 2. Materials and Methods

### 2.1. Bacterial Culture

For the experiment, an *E. coli DH5α* bacterial isolate (Thermo Fisher Scientific, Waltham, MA USA) was used; this isolate is widely used in cloning applications and has been shown to be highly efficient in transformation. Bacterial cultures were grown for 8 h in Luria broth at 37 °C and 220 rpm. *E. coli* was grown until the optical density of 0.5 was reached at a wavelength of 500 nm. Bacterial broth was concentrated until 1 OD. Bacterial cells were centrifuged (Velocity Minifuge 13μ, Dynamica Scientific Ltd., Livingston, UK) at 5000 rpm for 10 min and suspended in milk.

### 2.2. Media Preparation

Raw milk was obtained from local farmers, chilled on ice, and degassed before each PEF experiment. Vacuum degasification took place in a laboratory setting, for 1 h, at −0.8 bar, using a vacuum system consisting of a vacuum chamber and a vacuum pump (Mini Diaphragm Vacuum Pump N 816.3KN.18, KNF Neuberger GmbH, Freiburg, Germany). For comparison, six other pasteurized milk samples from local manufacturers were collected, differing in pasteurization techniques, such as low-temperature long-time (LTLT) at 63 °C for 30 min (Low-temperature long-time pasteurization line, Wenzhou Sijin Machinery Co., Ltd., Wenzhou, China), ultra-high temperature (UHT1) at 140 °C for 4 s (UHT pasteurization line, Shanghai Shangwang Machinery Manufacturing Co., Ltd., Shanghai, China), ultra-high temperature (UHT2) at 140 °C for 4 s (Tubular UHT milk pasteurization line, Shanghai Qingji Beverage Machinery Co., Ltd., Shanghai, China), ultra-high temperature (UHT3) at 140 °C for 3 s (UHT pasteurization line, Shanghai Beyond Machinery Co., Ltd., Shanghai, China), and microfiltration (MF1) using a 1.4 μm pore filter (Tetra Alcross M Bactocatch, Tetra Pak, Pully, Switzerland), followed by pasteurization at 76 °C for 20 s (Tetra Therm Lacta 1, Tetra Pak, Pully, Switzerland), microfiltration (MF2) using a 1.4 μm pore filter (Tetra Alcross M Bactocatch, Tetra Pak, Pully, Switzerland), followed by pasteurization at 76 °C for 30 s (Tetra Therm Lacta 1, Tetra Pak, Pully, Switzerland). Before every experiment, the conductivity of the milk samples was evaluated at pH 7 and was approximately 4.5 mS/m.

### 2.3. PEF Treatment

The PEF equipment consisted of an electrical pulse generator (BTX T 820, Holliston, MA, USA), a digital oscilloscope (Rigol DS2072A, Rigol Technologies Inc., Bedford, OH, USA), and a treatment chamber with two parallel electrodes. Electrodes were made of two equal polished stainless steel (AISI 304) plates, and a polymethylmethacrylate (PMMA) cube was used to isolate the electrodes, forming a treatment chamber with a gap of 0.1 cm. During the experiment, samples of milk suspension were treated with monopolar rectangular pulses in the range of 0.5 to 2.4 kV/cm with a varying number of pulses. The electric field strength was calculated according to Equation (1) [52]:

$$E = U/d, \tag{1}$$

where E is the electric field strength (V/cm), U is the voltage (V), and d is the distance between the electrodes (cm).

In all the experiments, the duration of a single pulse was 25 μs, and the pulse repetition rate was 1 Hz.

### 2.4. Dertermination of the Most Effective PEF Parameters

The experimental scheme is presented in Figure 1. After centrifugation of the concentrated *E. coli* suspension and removal of the eluent, the remaining bacteria were mixed with 1 mL pasteurized milk. Milk samples were treated using various combinations of PEF strengths and number of pulses. Immediately after PEF treatment, the suspension was removed from the cuvette and diluted with pasteurized milk at ratios of 1:100, 1:1000, and 1:10000. Then, 10 μL of the suspension was gently spread over an agar plate and incubated at 37 °C for 24 h. After incubation, the bacterial viability was evaluated by counting the number of colonies.

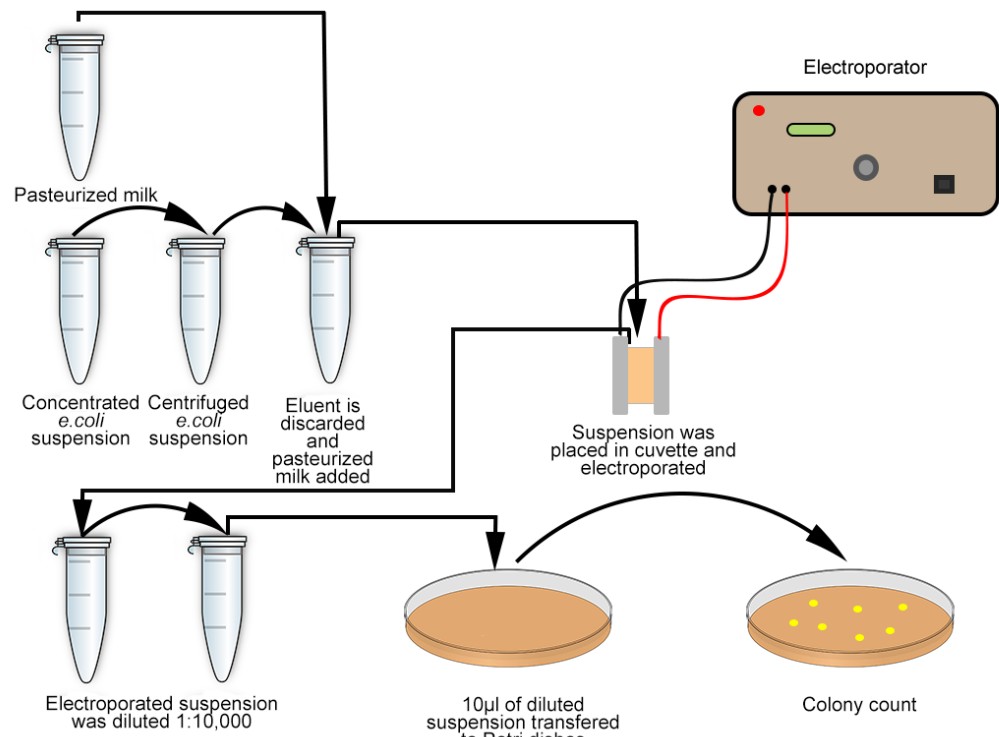

**Figure 1.** Schematics of an experiment to determine the most effective PEF parameters for bacterial inactivation in milk samples.

### 2.5. Determination of Bacterial Activity in Milk

During the experiment, we checked for three main parameters of microbial activity —in particular, coliforms, *L. monocytogenes*, and the total number of bacteria found in milk samples. While evaluating milk quality in the food industry, the coliforms and total bacteria count (during production) and *L. monocytogenes* (for the product) are usually checked to prove safety for consumption. The total bacterial count was determined on PCA agar after incubation for 72 h at 30 °C. Coliform count was determined on VRBL agar after incubation for 24 h at 30 °C with confirmation in BGLB broth. *L. monocytogenes* count was determined on AL agar (*Listeria* agar according to Ottaviani and Agosti medium) with confirmation of presumptive colonies. Confirmation tests for L. monocytogenes were conducted according to ISO 11290-2:2017, where L-Rhamnose +; D-Xylose -; beta-hemolysis +. Total bacterial count and coliforms were detected using a pour plate technique (1000 μL) according to ISO 4833-1:2013. L. monocytogenes were detected using a surface plate technique (100 μL) according to ISO 4833-1:2013. The experiments were performed according to the scheme presented in Figure 2.

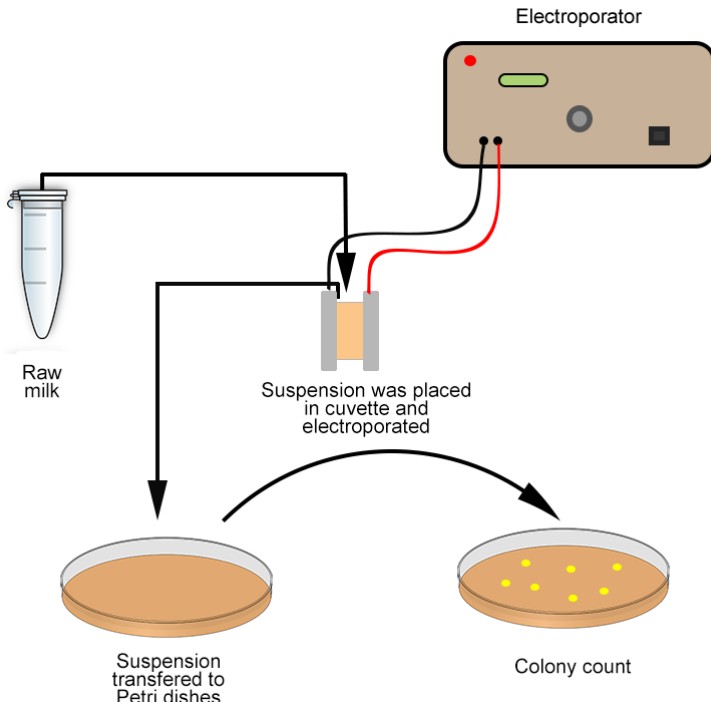

**Figure 2.** Schematics of an experiment to determine bacterial activity in raw milk samples after PEF treatment.

Raw milk samples obtained were placed in an electroporation cuvette and electroporated, 1000 µL suspension was transferred to Petri dishes, and a colony count was carried out.

### 2.6. Determination of Undenatured Whey Protein Content in Milk

Undenatured whey protein content was determined according to the method described by Kuramoto [54], where undenatured whey protein nitrogen was determined.

### 2.7. Determination of β-Lactoglobulin in Milk

β-Lactoglobulin content was determined using high-performance liquid chromatography. Standard solutions for the calibration curve were prepared from β-lactoglobulin in deionized water, at 10, 7.5, 5, and 2.5 mg/g concentrations. Milk samples were treated with HCl solution to obtain pH 4.6 (isoelectric point for proteins), and they were filtered through S&S filter paper and additionally through a 0.45 µm syringe filter.

The Shimadzu Prominence series (Shimadzu Corp., Kyoto, Japan) HPLC system with the TSKgel G2000 SWXL column (length 30 cm, internal diameter 0.78 cm) and TSKgel SWXL guard column (length 4 cm, internal diameter 0.6 cm) was used for separation and quantification of β-lactoglobulin. Mobile phase: buffer prepared dissolving 1.74 g $K_2HPO_4$, 12.37 g $KH_2PO_4$, and 21.41 g $Na_2SO_4$ in 1 L deionized water, pH 6.0. Prepared buffer was heated for 15 min in a water bath at 85 °C. Flow rate 1 mL/min, column temperature 30 °C, UV detector was set at 280 nm, injection volume 20 µL.

### 2.8. Temperature Measurements

The temperature of the milk media was measured using a UTi260B infrared thermal imager (Unit-Trend Technology Co., Ltd., Dongguan, China) with the temperature measurement accuracy of ±2 °C and a measuring range from −15 to 550 °C.

### 2.9. Statistics

All experimental conditions for the determination of optimal PEF parameters for *E. coli* inactivation, total bacterial count, *coliform* count, *L. monocytogenes* count, unde-

natured whey protein content and β-lactoglobulin content were maintained for three independent replicates (n = 3). One-way analysis of variance (ANOVA) and two-way ANOVA with Bonferroni analysis were performed. The Bonferroni test ($p < 0.05$) was considered statistically significant.

## 3. Results

To determine the most efficient PEF parameters for milk pasteurization, the *E. coli* suspension was prepared in milk and was exposed to various electric fields by varying the number of pulses and field strength (Figures 3 and 4). From these results, we selected 20 pulses, at 24 kV/cm pulse strength, and 25 μs of pulse duration as the optimal PEF parameters for further studies and for comparison with other pasteurization methods: LTLT, UHT, MF (Table 1).

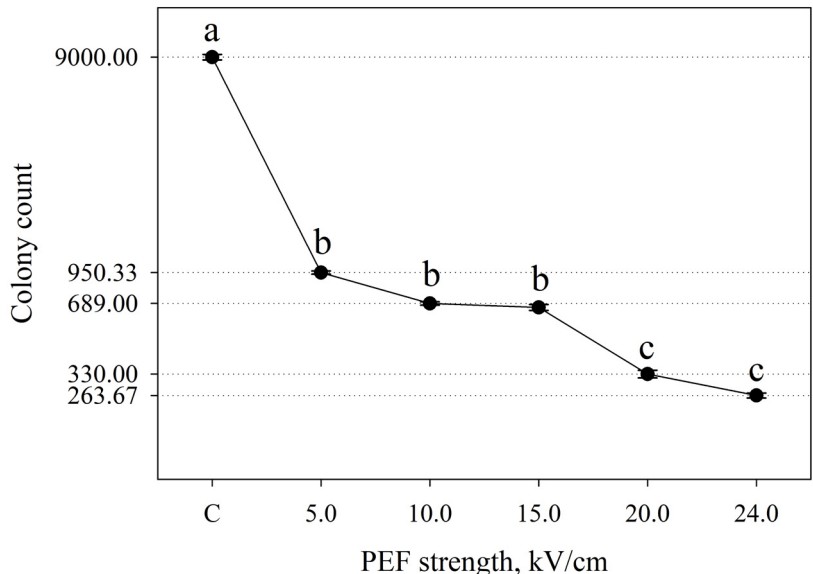

**Figure 3.** *E. coli DH5α* inactivation dependence on PEF strength, after 24 h of incubation in agar plates at 37 °C. The abscissa shows the voltage used during the experiment; the ordination axis presents the average number of colony count for each voltage. C—Control (raw milk). Control and PEF-treated suspensions were diluted with pasteurized (UHT) milk at ratio of 1:100. Different lowercase letters indicate significant differences between the samples ($p < 0.05$).

**Table 1.** Bacterial count in milk samples treated with different pasteurization techniques: RM—raw milk (control); PEF—pulsed electric field; LTLT—low-temperature long-time; UHT1—ultra-high temperature; UHT2—ultra-high temperature; UHT3—ultra-high temperature, MF1—microfiltration; MF2—microfiltration after 24 h of incubation in agar plates at 37 °C. Note: results are expressed as mean ± standard deviation (SD).

| No. | Milk Samples | Total Number of Bacteria Found in Milk Samples, cfu/mL | | Coliform Bacteria, cfu/mL | | *L. monocytogenes bacteria* |
|-----|--------------|-------|-------|-------|-------|-------|
| | | **Mean** | **SD** | **Mean** | **SD** | |
| 1. | RM | $33.33 \times 10^4$ | $5.77 \times 10^4$ | 1333.00 | 57.70 | Not detected |
| 2. | PEF | $4.43 \times 10^4$ | $0.35 \times 10^4$ | 5.00 | 1.00 | Not detected |
| 3. | Milk MF1 | 53.67 | 2.52 | <1 | - | Not detected |
| 4. | Milk MF2 | 15.00 | 2.00 | <1 | - | Not detected |
| 5. | Milk LTLT | 36.67 | 3.21 | <1 | - | Not detected |
| 6. | Milk UHT1 | <1 | - | <1 | - | Not detected |
| 7. | Milk UHT2 | <1 | - | <1 | - | Not detected |
| 8. | Milk UHT3 | <1 | - | <1 | - | Not detected |

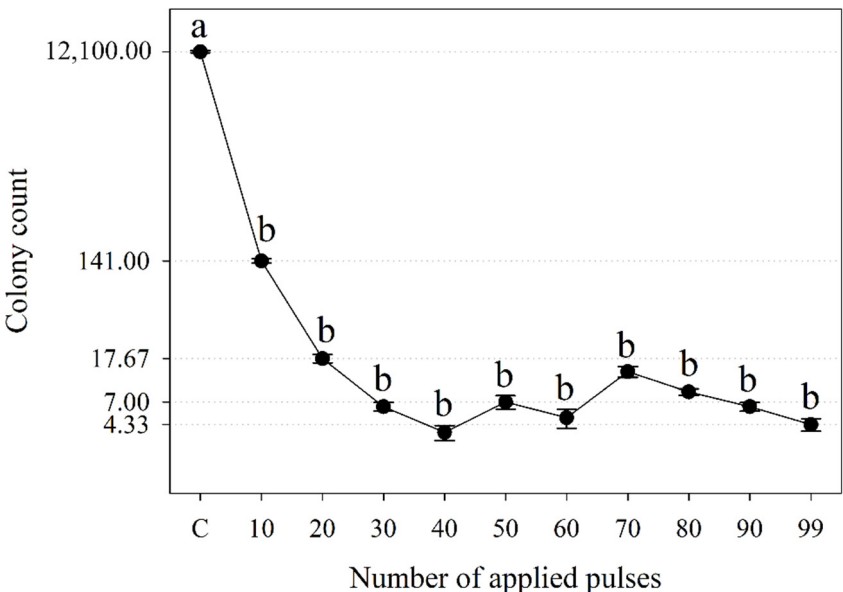

**Figure 4.** *E. coli DH5α* inactivation dependence on the number of pulses at 24 kV/cm voltage, after 24 h of incubation in agar plates at 37 °C. The abscissa shows the number of pulses used during the experiment; the ordination axis presents the mean of colony count. C—Control (raw milk). Control and PEF-treated suspensions were diluted with pasteurized (UHT) milk at ratio of 1:100. Different lowercase letters indicate significant differences between the samples ($p < 0.05$).

A significant statistical difference ($p < 0.05$) in bacterial inactivation was achieved by comparing the results obtained with the control (Figure 3). Bacterial inactivation at a PEF strength of 5 kV/cm was equal to 970 colonies, whereas the number of bacterial colonies in the control was 9000, which corresponded to $9 \times 10^7$ cfu/mL. Bacterial viability compared to the control decreased by 0.97 log. At pulse strengths of 10, 15, 20, and 24 kV/cm, the cfu values were 700, 680, 330, and 270, respectively, which corresponded to 1.11, 1.12, 1.44, and 1.52 log reductions in bacterial viability.

Milk samples were treated with 24 kV/cm field strength by increasing the number of pulses, and the colony count was assessed based on the number of dilutions made (Figure 4). A significant decrease in bacterial viability was achieved in the samples treated with 10 and 20 pulses. The colony count in the control yielded 12,100, which corresponds to $1.2 \times 10^8$ cfu/mL. The numbers of colonies at 10, 20, 30, 40, 50, 60, 70, 80, 90, 99 pulses were 140, 18, 7, 4, 5, 5, 6, 6, 6, 5, respectively. At these pulse numbers, bacterial viability compared to the control decreased by 1.94, 2.83, 3.24, 3.48, 3.38, 3.38, 3.30, 3.30, 3.38 log, respectively.

In samples treated with UHT, bacterial activity and *L. monocytogenes* were not detected. Raw milk samples were contaminated the most, with a total bacteria count of $33.5 \times 10^4$ cfu/mL and coliforms of 1333 cfu/mL. Comparing raw milk samples to treatment with PEF, we measured a significant decrease in coliform count at 2.4 log, while the total bacterial count decreased by 0.9 log. In the samples treated with MF and LTLT pasteurization techniques, coliform bacteria were not detected, and LTLT technology reduced the total bacterial count by 3.96 log, while 3.79 and 4.35 log reductions were observed in samples treated with MF1 and MF2.

Undenatured whey protein content in raw milk samples and those treated with PEF was found to be the same; in both cases, it was 4.98 mg/mL (Figure 5). Milk samples treated with MF yielded 4.66 and 4.55 mg/mL undenatured whey protein, even though, after filtration, milk was pasteurized at 76 °C for 20 s (MF1) and 76 °C for 30 s (MF2). A significant difference in undenatured whey protein content was observed in samples treated with UHT. Samples yielded only 0.26, 0.19, and 0.51 mg/mL of undenatured whey protein, due to high treatment temperatures of 140 °C, at which whey proteins denaturize.

LTLT pasteurization yielded 1.85 mg/mL undenatured whey protein content because of the low temperatures used during processing at 63 °C for 30 min (Figure 5).

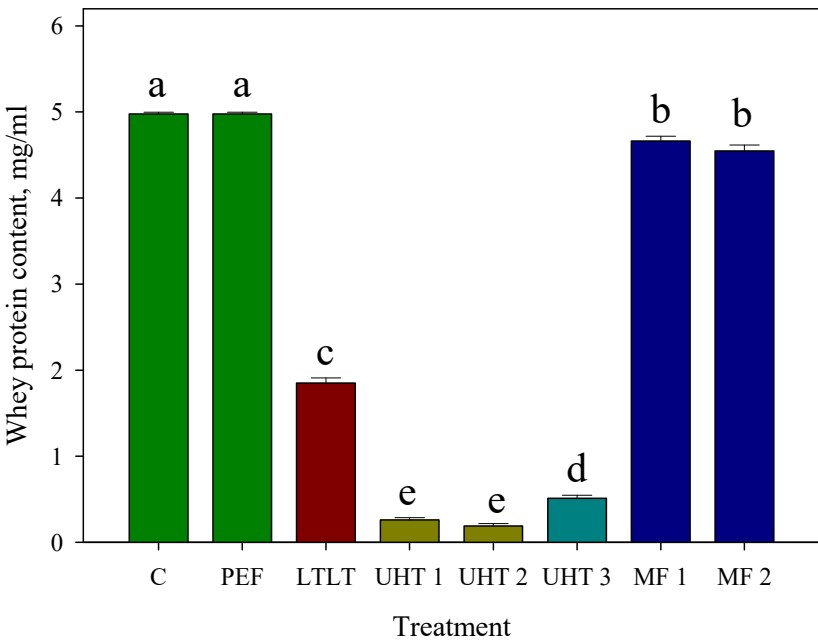

**Figure 5.** Undenatured whey protein content in milk samples treated with different pasteurization techniques: C—control (raw milk); PEF—pulsed electric field; LTLT—low-temperature long-time; UHT1—ultra-high temperature; UHT2—ultra-high temperature; UHT3—ultra-high temperature; MF1—microfiltration; MF2—microfiltration. Different lowercase letters indicate significant differences between the samples ($p < 0.05$).

After the effect of different pasteurization techniques on undenatured whey protein content was determined, experiments were performed to understand the effect of high-intensity electric pulses on β-lactoglobulin concentration, compared to traditional pasteurization techniques. Using HPLC, it was determined that PEF had no effect on β-lactoglobulin content and remained the same in the raw milk samples at 3.28 mg/mL (Figures 6a and 7).

Similar β-lactoglobulin concentration results were achieved in samples treated with microfiltration. During microfiltration, the filtrate was treated with low-temperature pasteurization for a short period of time. These treatment conditions had only a negligible effect on the β-lactoglobulin concentration, which remained relatively high for MF1 at 2.96 mg/mL and MF2 at 2.41 mg/mL, respectively (Figures 6c and 7). For representation purposes, the β-lactoglobulin standards and samples treated with pulsed PEF are presented in Figure 6b. Milk samples treated with UHT showed the most significant decrease in β-lactoglobulin concentrations, which were not detectable. The HPLC results of the milk samples treated with PEF and UHT are presented in Figure 6d. Milk samples treated with the LTLT pasteurization technique yielded 0.83 mg/mL β-lactoglobulin content (Figure 7). Although relatively high temperatures were not generated at 63 °C, prolonged exposure for 30 min had a significant effect on protein structure, which corresponded to a 75% decrease in β-lactoglobulin content compared to raw milk samples. Several reports state that heat treatment at temperatures above 60 °C initiates the unfolding of the globular structure of whey proteins, which results in protein denaturation [55,56], and a long treatment time of 30 min may result in a decrease in whey protein content during pasteurization.

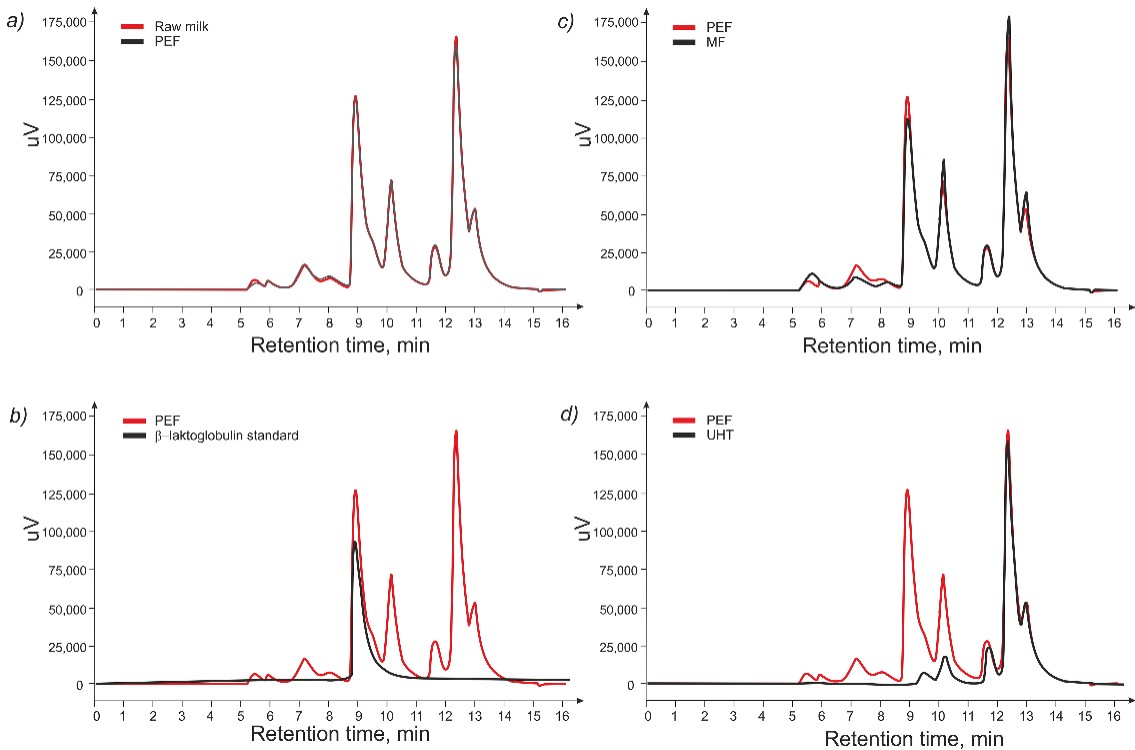

**Figure 6.** HPLC results on β-lactoglobulin content in raw milk samples and those treated with PEF, (**a**); HPLC results on β-lactoglobulin content in PEF-treated samples and β-lactoglobulin standard, (**b**); HPLC results on β-lactoglobulin content in milk samples treated with MF and PEF, (**c**); HPLC results on β-lactoglobulin content in milk samples treated with UHT and PEF, (**d**).

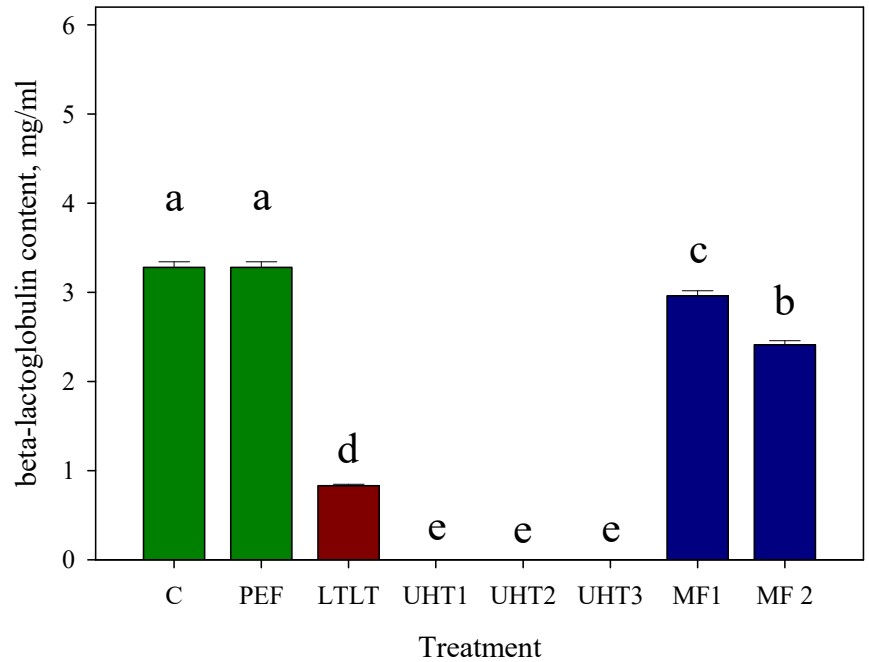

**Figure 7.** β-lactoglobulin content in milk samples treated with different pasteurization techniques: C—control (raw milk); PEF—pulsed electric field; LTLT—low-temperature long-time; UHT1—ultra-high temperature; UHT2—ultra-high temperature; UHT3—ultra-high temperature; MF1—microfiltration; MF2—microfiltration. Different lowercase letters indicate significant differences between the samples ($p < 0.05$).

## 4. Discussion

### 4.1. Differences in Bacterial Inactivation

In this study, we aimed to examine the effect of PEF on whey protein content and bacterial inactivation. The results were compared to traditional pasteurization techniques used in the food industry. PEF results on whey protein content obtained in this study were comparable with those reported in previous publications [57,58]. It has been reported that, while achieving significant bacterial inactivation by PEF, it has minimal to no effect on whey protein content, compared to control samples [57]. We were able to confirm this, as can be seen in Figures 5 and 7. However, a report published on how PEF affects the structural properties of whey protein isolates contradicts this opinion [59]. It has been reported that the PEF intensity and the number of pulses play a major role in the structural changes of whey protein isolates [59]. In this study, we did not test PEF's effect on the structural properties of WP, so we cannot draw any conclusions regarding this. It has been widely suggested that increasing the electric field strength and the number of pulses has a significant effect on bacterial inactivation [23,57,60–62]; we were able to achieve similar results in this study, as can be seen in Figures 3 and 4. The results in Table 1 show that the effect of PEF differs for individual species of bacteria. A more significant decrease in coliform numbers was achieved, and other bacteria might be more resistant to PEF because of the differences in bacterial morphology. It has been shown that bacterial shape, size, and cell wall thickness have significant effects on the efficiency of PEF [63]. Larger bacteria are more sensitive to PEF treatment because of the higher transmembrane potential induced during treatment. Moreover, bacteria with thicker cell walls are less sensitive to PEF-induced inactivation [63]. A wide variety of Gram-positive and Gram-negative bacterial species are found in raw milk, resulting in a different distribution between the total number of inactivated bacteria and coliforms [64]. Gram-negative bacteria seem to be more sensitive to electric pulses than Gram-positive bacteria. This difference most probably is related to differences in the composition and thickness of the bacterial cell wall [65–67]. Gram-negative bacteria have a thin layer of peptidoglycan and an outer membrane containing lipopolysaccharide, as opposed to Gram-positive. According to reports published previously, Gram-negative bacteria are more effective in performing electro-transformation than Gram-positive [68]. This is likely because of the bacterial cell envelope and cell wall, which reduces the amount of exogenous DNA entering the cell. According to articles published so far, bacterial inactivation by PEF occurs due to the breakdown of the cell membrane, affected by high-intensity electric fields [69–73]. Strong electric pulses permeate cells, causing damage to the cell membrane due to transmembrane potential. When this potential exceeds a critical value, the membrane becomes permeable to the outside medium [69–73]. The damage can be reversible or irreversible and depends on the electric field strength, duration, and number of pulses. Irreversible electroporation leads to damage to the cell membrane, which causes dielectric breakdown and the release of cell content [74–77]. The thick peptidoglycan layer might protect cells from lysis by preventing cell leakage from the cell. It has been shown that bacteria can modify peptidoglycan backbones to acquire resistance against antimicrobials [78]. Gram-negative and Gram-positive bacteria, and even different strains within these categories, react differently to pH, temperature, PEF, and different combinations of the described parameters [79,80].

### 4.2. Downside of Microfiltration Technology

Microfiltration is a practical method for maintaining whey protein content that has already been applied in the industry, but it has technical drawbacks. Ceramic filters tend to clog and constantly need to be regenerated or replaced. Because 1.4 μm pore filters cannot effectively filter out all the pathogenic bacteria and spores, the filtrate must be additionally treated with low-temperature (at 76 °C for 20 s or 30 s) pasteurization. Spores such as *B. licheniformis* can pass through 1.2 μm pore size filters [81]. In comparison, *E. coli*, *Sphingopyxis alaskensis*, *Brevundimonas diminuta*, *Vibrio cholera*, *Legionella pneumophila*, *Hylemonella gracilis spirillum*, and *Hylemonella gracilis* can pass through 0.45 and even

0.22 μm filters [82]. Heat treatment increases energy costs because the entire volume of the filtrate must be brought to a temperature of 76 °C. In contrast, during PEF treatment, the amount of heat released is minimal, undenatured whey protein content remains the same as in raw milk, and the technology can be incorporated into existing milk processing lines with a flow through PEF chambers. Microfiltration combined with pre-treatment with high-intensity electric pulses at 24 kV/cm, a pulse duration of 25 μs, and a number of pulses equal to 20 could be a new technology that provides lower raw milk pasteurization and operation costs while ensuring higher nutritive value in milk products.

### 4.3. Whey Protein Content Dependency

Results from the WPN experiments (Figure 5) showed that milk samples treated with high-intensity electric pulses had the same undenatured whey protein concentration as raw milk samples. This shows that, during the PEF treatment, at a 25 μs pulse length, high heat was not generated. Whey proteins, especially β-lactoglobulin, start denaturing at 70 °C [83], and when the temperature exceeds 115 °C, the extent of denaturation exceeds 90% [55]; however, the WPN method has several drawbacks. The main issue with whey protein nitrogen index (WPNI) tests is their poor reproducibility due to variable and unstable turbidity [84]; consequently, HPLC using a β-lactoglobulin standard was implemented. The α-lactoglobulin content was not measured during the experiments because it is more stable at high temperatures [55] and constitutes only 20% of whey protein, while β-lactoglobulin amounts to 50% [46]. It can be safely assumed that high-intensity electric pulses with the chosen parameters should not affect α-lactoglobulin content as protein denatures less at high pasteurization temperatures compared to β-lactoglobulin. During the PEF treatment, the temperatures generated with the chosen parameters were below 30 °C.

### 4.4. PEF Treatment Optimization

Bacterial inactivation could be further improved by increasing the number of applied pulses, from 20 to 40 pulses. Bacterial colony count averages at 40 pulses and a further increase in the number of pulse counts did not have a significant effect on bacterial inactivation, as can be seen in Figure 4. Another option would be to increase the pulse duration [85–87]. Further experiments are required to determine the optimal pulse duration. Nevertheless, the issue of spores remains, which could be overcome by designing pressurized systems. However, spores are inactivated at high pressure (>300 MPa) [88–90], and further research is required for the adaptation of PEF in liquid food processing lines.

### 4.5. PEF in Production of Whey Protein Powder

Pulsed electric fields could be a viable method for the production of whey protein powder. Milk treated with PEF has a higher concentration of undenatured whey protein. Raw milk processed with PEF could yield a higher concentration of raw whey protein for the production of whey protein isolates, concentrates, and hydrolysates for the sports industry or as a protein supplement for people suffering from Phenylketonuria.

### 4.6. Potential Limitations of the Study

The current study has several limitations. First, the apparatus used could generate electric fields only up to 3 kV; thus, we were not able to explore PEF parameters above 3 kV. In order to increase the electric field strength, a treatment chamber with a gap of 0.1 cm was used, which allowed us to achieve a PEF strength of up to 30 kV/cm. Nevertheless, because of the electric discharge occurring above 24 kV/cm, we were not able to use a higher strength. Another limitation of this study was the bacterial load in the raw milk samples, which was related to variations in milking and storing conditions. During this study, we were able to obtain reasonably clean samples with no *Listeria monocytogenes* present. Nevertheless, another issue was the variation in whey protein content in milk samples. It has been reported that whey protein content in raw milk may fluctuate due to seasonality [91–94]. In this study, the raw milk samples for the experiments were collected

during wintertime. Therefore, the whey protein concentration may have been significantly lower than that in milk samples obtained during summertime.

## 5. Conclusions

In this study, we demonstrated that PEF can significantly reduce the bacterial count in treated samples under the following conditions: PEF at 24 kV/cm, pulse duration 25 μs, 20 pulses. We observed reductions of 2.43 and 0.9 logs in coliforms and total bacteria count, respectively. Notably, under these PEF conditions, undenatured whey protein content remained unchanged in comparison to the raw milk samples at 4.98 mg/mL. β-Lactoglobulin content was not affected, yielding the same results as in the raw milk samples, at 3.28 mg/mL. Regarding milk samples treated with traditional pasteurization techniques, MF yielded 4.66 and 4.55 mg/mL, UHT yielded 0.26, 0.19, and 0.51 mg/mL, and LTLT yielded 1.85 mg/mL of undenatured whey protein content. The β-lactoglobulin concentration results of milk samples treated with MF remained relatively high at 2.96 mg/mL and 2.41 mg/mL, LTLT yielded 0.83 mg/mL, and in samples treated with UHT, β-lactoglobulin was not detected. Coliforms were not observed in samples treated with traditional pasteurization techniques, while the total bacterial count in MF-treated milk yielded 3.79 and 4.35 log reductions and LTLT by 3.96 log; in UHT-treated milk, bacterial activity was not observed.

**Author Contributions:** Conceptualization, A.Š. and S.Š.; methodology, P.R.; software, D.U.; validation, A.Š., P.R. and S.Š.; formal analysis, D.U.; investigation, A.Š.; resources, A.Š.; data curation, M.V.; writing—original draft preparation, A.Š.; writing—review and editing, A.Š.; visualization, M.V.; supervision, S.Š.; project administration, A.Š.; funding acquisition, S.Š. All authors have read and agreed to the published version of the manuscript.

**Funding:** This research funded by local Vytautas Magnus University fund.

**Institutional Review Board Statement:** Not applicable.

**Informed Consent Statement:** Not applicable.

**Acknowledgments:** The authors thank the Food Institute of the Kaunas University of Technology for technological support.

**Conflicts of Interest:** The authors declare no conflict of interest.

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
