# Peer review of "Effect of Pulsed Electric Field (PEF) on Bacterial Viability and Whey Protein in the Processing of Raw Milk"

_applsci, doi:10.3390/app112311281_

Round 1

Reviewer 1 Report

Introduction

The first lines have a journalistic style, there is no need to include them. Please start from: Thermal pasteurization of food products and beverages has specific drawbacks, it reads better for a scientific paper.

PEF: please explain the abbreviation the first time mentioned in the main text.

Please divide the introduction in two subsections to facilitate future readers – no need to shorten, it is ok for length and content.

Please mention clearly the hypothesis upon which the work had been based.

Materials and methods

2.1. Please present history and details of this isolate (even in supplementary material). Also, use isolate rather than strain.

2.2. degassed? Please specifically cow milk.

2.4. For TBC, can you please confirm that you followed the recommendations of the APHA? Thank you.

Listeria: Use of Ottaviani agar only poses some problems regarding correct identification. There are many references describing the correct procedure and a relevant ISO has been developed. However, I assume that this work did not include these procedures. This is a limitation and please mention accordingly in the discussion.

2.8. Please explain how you handled the triplicate findings for analysis.

Results

Please start Results by providing some information regarding variation between the triplicates performed during the study.

Still to note the comment regarding the Listeria detection method.

Figures 3 and 5, please use colours to make better for readers.

Discussion

Please add a new subsection 4.6. Potential limitations of the study to include the various issues as above.

In general, the manuscript is suitable for the journal and can advance to revision. After careful revision by the authors by taking into account all the points above, re-evaluation will be necessary.

Author Response

Please find in file

Reviewer 2 Report

Dear Author(s)

After an exhaustive revision, the manuscript is Reconsider after major revision (control missing in some experiments). In general, the study is closely connected to the journal's objectives. The study is very interesting. The English is good. The introduction is complete, very detailed, but is a lot (excessive) of information. The section of materials and methods is very complete, but the authors need some references and a Figure. Moreover, the authors need to make changes and modify parts in the manuscript, mainly in the section “Results and Discussion”, since it is the more incomplete subsection.

In the following pages, I give a detailed revision of the manuscript.

Best regards

Affiliations

The authors must add all the initials and emails of the authors in the affiliation section.

ABSTRACT

The abstract is good. However, the authors need to add more numerical results.

  1. INTRODUCTION

The introduction is very clear, with good English, and it presents references from 2021. I consider that the introduction is very long; the authors should reduce the introduction, and focus on the most important. The examples should be more focused on the matrices used in the study, and the other examples could be used as part of results and discussion.

  1. MATERIALS AND METHODS

General comments

This section is clear. The English is good. The authors must add a Figure that represents all the methodology in the section Materials and Methods. This Figure will help to understand the methodology. Some observations:

2.4. Determination of bacterial activity in milk

What is the reference?

2.6. Determination of β-Lactoglobulin in milk

What is the reference?

  1. RESULTS

"Results" is characterized by a description of the results.

This section is very complete. However, need more numerical data in the description of results on the Figure 1.

  1. DISCUSSIONS

"Discussions" is characterized by the explication of the results, comparison with other studies, and explication (discussion) of the results obtained with respect to other studies. My observations:

4.1. Differences in bacterial inactivation

4.2. Downside of Microfiltration technology

4.3. Whey protein content dependency

4.4. PEF treatment optimization

4.5. PEF in production of whey protein powder

All the subsections are very complete in terms of the explication of the results. However, the authors need to add the comparison with other studies, and explication (discussion) of the results obtained with respect to other studies.

  1. Conclusions

The conclusions are good, and it has concordance with the results.

Author Response

Please find in file

Reviewer 3 Report

Dear Authors,

The paper authored by Šalaševičius et al. is interesting and represents a very useful contribution to increase of knowledge in this field. However, there are English errors that if not corrected will detract from your interesting paper. This also applies to the length of sentences, some are too long, which make the article difficult to read at times. There are many commas in the text that are used unjustifiably. Also, please don't use this many semicolons, combine sentences with each other more neatly. The text has to undergo English editing in terms of grammar and  punctuation. Mixing of tenses also occurs in the paper. Also, some improvements are required in the chapter Materials and methods, because the authors did not do their best to accurately describe the methods and equipment used. I guess these shortcomings can be fixed quickly. This is why, I recommend this article to be published in Applied Sciences  with minor revision.

Below you will find a list of detailed comments.

Abstract: There is growing….

Line 9 - …applied in the industry. However, food products…

Line  16 - Please do not use the term “ high drop” here and in the paper as it is not the scientific language, use significant decrease/decrement instead.

Line 18 – whey protein content was affected / it was detrimental for whey protein content

Line 19 - …treated with UHT. LTLT treatment introduced a significant increase of…

Line 27 - …resulted in deaths – this sentence is actually a great example of improper mixing of tenses. And there are many like this

Line 34 – An inability to ensure daily….

Line 51 – effective in

Line 54 - Please start new sentence - …food processing lines. In comparison…

Line 64 - …cortex. Therefore, ….

Line 68 – Raw whey protein? Just one protein? Do you mean specific fraction in particular? I think you should go plural and state “Raw whey proteins in milk are highly prized owing to their nutritional value”

Line 69 – please correct to whey protein preparations

Line 70 – please cross out “ powdered” and change to whey protein isolates

Line 71 – Whey protein isolates and concentrates are highly adopted….

Speaking of nutrition in sports, there is a lack of current publications on special product fortification technologies with whey protein preparations. Please cite the following papers to fill the gap. 

Nastaj et al. (2020). Effect of erythritol on physicochemical properties of reformulated high protein meringues obtained from whey protein isolate.

Nastaj et al. (2019). Physicochemical properties of High-Protein-Set Yoghurts obtained with the addition of whey protein preparations

Line 72 – Whey proteins have been shown…

Line 76 and 79 – once you write Phenyloketonuria  with an uppercase letter and  in line 79 with a lowercase letter

Line 79- where is the reference number?

Line 80 – Whey proteins mostly consist of

Line 82 – …and other amino acids, which stimulate…

Line 87 - …and their nutritional content are low

Line 89 – start a new paragraph with indentation

Pulsed electric field can bridge the gap ….

Lines 91 and 98 – pulse electric fields? or field ?

Use the  PEF abbreviation wherever possible

Line 103 – please cross out growing medium

Line 106- optical density was reached

Line 107 – please define the apparatus plus details (location)

Line 109 – what milk degasser did the farmers use? Give all the details

Paragraph 2.2 - Please define the equipment for the thermal process of milk

Line 111 – differing in pasteurization techniques

Line 114 – and its approximate value was 4.5 mS/m

Line 117/118 – please give the exact location in the USA, city, state?

Line 123 – The electric field strength was calculated from the equation [52]:

Line 142 - …according to Ottaviani and Agost [please add reference number in the brackets]

Line 148 - please define the apparatus plus details (location)

Besides, I believe that a comparative study of chromatographic profiles was carried out based on the masses of standard proteins, please provide all the details

Line 154 - with the temperature measurement accuracy of ±2 °C

line 154 infrared thermal imager – location plus details

Please combine  RESULTS and DISCUSSION in your paper, this will make it easier to read and you won't have to repeat the same content

Lines 166-180– sentences out of place and should be moved to materials and methods section

 Line 202 -A significant decrease in…

Lines 204-206 – please reword the sentence

Please redo Figure 3, resolution is too low and I see the pixels:)

Lines 225-227 – please reword / …at which whey proteins denaturize

Please enlarge and redo Figure 4, it is not very clear

Line 235-236 – this sentence does not belong here

Please redo Figure 5, resolution is too low and I see the pixels:)

Line 245-249 -sentence too long and needs rewording / instead of little effect use insignificant or neglible effect

251- redefine drop

Line 258 -globular structure of whey proteins

Line 263 -the results from table 1 show - please rearrange the figures and tables so that they correspond to the actual text that describes them. It will make yore paper more reader friendly

Line 264 - redefine drop

Lines 268-277 – please don’t mix tenses

Lines 277-282 – sentence too long and needs rewording / effective in performing

Line 286- The damage can be reversible…

Lines 295-297 – please reword the sentence

Line 297 – The filters tend to clog up

Lines 298-304– sentence too long and needs rewording

Line 313- please remove the entire sentence

Lines 314-316- please do not mix tenses

Line 317 – Whey proteins,….

Line 335 – bacterial kill count? – please use the scientific language

Line 339 – However, spores

Lines 343-351 – these sentences match the conclusions /line 349 – whey protein isolates, concentrates…

I know that was not the purpose of the study, but can you determine basing on the literature, how PEF will affect other milk components: lactose, casein, fat, vitamins and minerals? I think these  informations would be interesting to the readers at the end

Good luck with the corrections:)

Author Response

Please find in file

Round 2

Reviewer 1 Report

The authors have provide plausible answers to the comments and have made acceptable changes in the revised manuscript.

The manuscript has been improved.

I have no further comments in the scientific aspects, as they authors have addressed all previous comments successfully.
However, there is a need for detailed evaluation of linguistic aspects of the manuscript. I suggest that the authors revise carefully and correct the entire  manuscript for language problems.
The corrected version can be assessed by the academic editor.
Hence, minor revision.

Author Response

Following your suggestion to improve English language, we used MDPI service for English editing. Please find the English revised version of the manuscript. 

Thank you for you efforts that helped us to improve the manuscript.

Reviewer 2 Report

Dear Author(s)

After an exhaustive revision, the manuscript is Accept in present form. The resubmitted manuscript has been completely improved compared to its previous version. Therefore, the manuscript can be published in “Applied Sciences”.

Best regards.

Author Response

Thank you for you efforts and time that helped us to improve the manuscript